# Profiling the colonic mucosal response to fecal microbiota transplantation identifies a role for GBP5 in colitis in humans and mice

Laurence D. W. Luu [1,2,12], Abhimanu Pandey [3,12],
Sudarshan Paramsothy [4,5,12], Chinh Ngo[3], Natalia Castaño-Rodríguez [2],
Cheng Liu[6,7,8], Michael A. Kamm[9,10], Thomas J. Borody [11], Si Ming Man [3] &
Nadeem O. Kaakoush [1] ✉

Host molecular responses to fecal microbiota transplantation (FMT) in ulcerative colitis are not well understood. Here, we profile the human colonic mucosal transcriptome prior to and following FMT or placebo to identify molecules regulated during disease remission. FMT alters the transcriptome above the effect of placebo (n = 75 vs 3 genes, q < 0.05), including modulation of structural, metabolic and inflammatory pathways. This response is attributed to responders with no consistency observed in non-responders. Regulated pathways in responders include tight junctions, calcium signalling and xeno-biotic metabolism. Genes significantly regulated longitudinally in responders post-FMT could discriminate them from responders and non-responders at baseline and non-responders post-FMT, with *GBP5* and *IRF4* downregulation being associated with remission. Female mice with a deletion of *GBP5* are more resistant to developing colitis than their wild-type littermates, showing higher colonic IRF4 phosphorylation. The colonic mucosal response discriminates UC remission following FMT, with GBP5 playing a detrimental role in colitis.

The complex etiology of ulcerative colitis (UC), one of the major subtypes of inflammatory bowel diseases (IBD), involves host genetic and immune factors, environmental factors, and the gut microbiome[1]. While there is currently no cure for UC, active disease can be controlled and treated effectively through either 5-aminosalicylic acid derivatives or immunosuppression with steroids and/or biologics. However, patients can be unresponsive to these therapies and those on them for long periods can have adverse effects or develop resistance[2,3].

Therapies to manipulate the microbiota beyond antibiotics have been trialed for the treatment of UC. This includes whole microbial communities in the form of fecal microbiota transplantation (FMT) from healthy donors[4–10] or more targeted live microbial therapeutics[11]. Current evidence indicates that FMT is effective in UC[12–16], albeit not all patients respond positively. Further, in those who respond, duration is limited and disease relapses[17], with maintenance therapy required to sustain benefit[18–20]. These findings have led to concerted efforts to assess for biomarkers of efficacy[19,21,22], with emphasis on bacterial biomarkers that could be developed into defined therapeutic microbial consortia.

Less studied are the consequences of FMT on the host mucosal microenvironment and the differences in the host response relative to

[1]School of Biomedical Sciences, UNSW, Sydney, NSW 2052, Australia. [2]School of Biotechnology and Biomolecular Sciences, UNSW, Sydney, NSW 2052, Australia. [3]The John Curtin School of Medical Research, The Australian National University, Canberra, ACT 2601, Australia. [4]Concord Clinical School, The University of Sydney, Sydney, NSW 2139, Australia. [5]Department of Gastroenterology, Concord Repatriation General Hospital, Sydney, NSW 2139, Australia. [6]Conjoint Gastroenterology Laboratory, QIMR Berghofer Medical Research Institute, Herston, QLD QLD, Australia. [7]School of Medicine, University of Queensland, Herston, QLD 4006, Australia. [8]Mater Pathology, Mater Hospital Brisbane, South Brisbane, QLD 4101, Australia. [9]Department of Gastroenterology, St Vincent's Hospital, Melbourne, VIC 3065, Australia. [10]Department of Medicine, University of Melbourne, Melbourne, VIC 3010, Australia. [11]Centre for Digestive Diseases, Sydney, NSW 2046, Australia. [12]These authors contributed equally: Laurence D. W. Luu, Abhimanu Pandey, Sudarshan Paramsothy. ✉e-mail: n.kaakoush@unsw.edu.au

clinical efficacy. There is some evidence in animal models that specific host factors can be influential in the response to FMT. For example, interleukin (IL)−33 has been found to play a key role in protecting against disease resulting from *Clostridioides difficile* infection through the activation of type-2 innate lymphoid cells, and repopulation of the gut microbiota through FMT can rescue the levels of colonic IL-33[23]. The anti-inflammatory cytokine IL-10 also appears to play a role as IL-10 receptor blockade during FMT led to decreased efficacy during chemically induced colitis[24]. Hence, examining the effect of FMT on the host in response or lack thereof would provide crucial information on the microenvironment in disease remission, but could also potentially enable the identification of host factors associated with FMT mechanisms of action.

Here, we characterize the host response in UC following FMT while accounting for treatment efficacy, through assessment of the mucosal transcriptome of patients recruited into our previously published clinical trial. We validate the role of one gene in development of colitis through a loss-of-function mouse model followed by induction of experimental colitis.

## Results

### Characteristics of sequencing data in study cohort
A subset of patients was selected on the basis of matching for therapy outcome ($n = 13$ reached the primary trial endpoint; $n = 13$ did not) (Fig. 1A). Sixty RNA samples derived from these patient mucosal biopsies were depleted from ribosomal RNA and shotgun sequenced. This corresponded to patient baseline samples ($n = 26$) prior to FMT therapy, and their samples after FMT ($n = 26$), as well as eight samples from patients that were initially on placebo (Fig. 1A). While the microbiota changes in cohort have been previously extensively characterized[8,25], FMT was confirmed to have had a significant impact on the alpha and beta diversity of the colonic mucosal microbiota in the selected patients (Supplementary Fig. 1A, B).

Raw transcriptome data across the 60 samples were homogenous (Supplementary Fig. 2A) with an average read depth per sample of $79235607 \pm 1774103$ reads (±SEM; standard deviation = 13742146; Supplementary Data 1). Counts of expressed transcripts per sample were also found to be consistent across treatment groups (Supplementary Fig. 2B). Estimation of cell type abundances using CIBERSORTx found no significant differences across sample groups within the bulk RNAseq data (Supplementary Fig. 2C), inferring that differential transcript expression was not a consequence of histological differences in overall cellular composition of the biopsies. Notably, analysis of the specific cell types suggested differences in the inferred abundances of naïve ($p = 0.065$) and memory CD4 T cells (resting, $p = 0.0095$; activated, $p = 0.05$) (Supplementary Fig. 2D), which is in line with CD4[+] but not CD8[+] T cells playing a role in FMT-mediated resolution of *C. difficile* infection[26].

### Effect of FMT on host mucosal expression in patients with UC
The longitudinal effect of FMT on patient gut mucosal expression (Tx0 vs Tx8) was assessed and compared to the effect of placebo (Tx0 vs P8). 75 transcripts were found to be differentially expressed with FMT ($q < 0.05$; 276 transcripts at $q < 0.1$) as compared to three transcripts with placebo ($q < 0.05$; 3 transcripts at $q < 0.1$) (Fig. 1B, D; Supplementary Data 2,3). To account for differences in power between the comparisons ($n = 26$ patients for FMT and $n = 8$ for placebo), pathway enrichment were performed with GAGE, which analyzes regulation across the whole transcriptome as opposed to only significantly regulated genes. This confirmed substantial differences in the effects of FMT and placebo (Fig. 1C, E), with 49 pathways significantly regulated with FMT compared to 12 pathways with placebo ($q < 0.05$). Relevant pathways found to be regulated with FMT and not placebo include tight junctions, TGF-β signaling, sphingolipid metabolism, regulation of the actin cytoskeleton, MAPK

signaling, lysosome, leukocyte transendothelial migration, and bile secretion, among several others (Fig. 1C, E, F), showing that FMT had a clear impact on a patient's colonic mucosa beyond the effect of placebo.

### Host signatures associated with disease remission in patients with UC
The effect of FMT was then investigated in the context of disease remission, whereby the longitudinal changes in expression in responders (Tx0Y vs Tx8Y) and non-responders (Tx0N vs Tx8N) were identified. A clear finding from this was that the observed changes with FMT therapy were driven by responders, with 78 transcripts significantly differentially expressed in responders as compared to none in non-responders (Fig. 2A, B). This dramatic difference was consistent when the q-value was relaxed to 0.1 (responders: 140 transcripts; non-responders: 0 transcripts; Supplementary Data 4). It was even more evident when a paired analysis by patient was performed (responders: 1207 transcripts and non-responders: 1 transcript at $q < 0.05$; Supplementary Data 5, 6). This again was highly consistent on pathway analysis with GAGE, with 10 pathways significantly upregulated in responders as compared to 1 pathway significantly downregulated in non-responders (Fig. 2C, D). Of all the pathways, tight junctions and calcium signaling were of particular interest as they were a consistent result in FMT (Fig. 1C, F) and not in placebo (Fig. 1E), and had a substantial number of genes attributed to them (Fig. 2E). Intriguingly, while responders had higher baseline alpha diversity (Supplementary Fig. 3A, B), and a significantly different beta diversity (Supplementary Fig. 3C), the microbial differences between responders and non-responders were not as pronounced as the host mucosal response. It is also important to note that no genes were identified as significantly regulated after correction for FDR when responders were compared to non-responders at baseline (Tx0Y vs Tx0N).

In addition to differential gene expression, differences across transcript splicing were examined. 100 splicing events (8 A3SS, 8 A5SS, 15 MXE, 4 RI, and 65 SE) and 98 events (17 A3SS, 6 A5SS, 8 MXE, 5 RI, and 62 SE) were identified to significantly change with FMT in responders and non-responders, respectively (Supplementary Fig. 4A, B, Supplementary Data 7–9). Despite the similar number of events across responders and non-responders, there appeared to be more consistency in the nature of the events in responders, with KEGG pathway analysis identifying 4 significantly enriched pathways ($q < 0.05$) in genes with splicing events in responders compared to none in non-responders (Supplementary Fig. 4C). Of the genes contributing to pathways, the exon skipping event within spleen associated tyrosine kinase *SYK* was the most differentially changed in responders (Supplementary Data 8), which is of interest considering it is involved in antifungal immunity and inflammasome activation[27]. The impact of this skipping event at the protein level was inferred, and this suggested that different levels of long and short forms of SYK were present in responders and non-responders (Supplementary Fig. 4D).

The results indicated that patients that achieve remission following FMT have a unique mucosal response that is not replicated in those that do not, with a relevant role for genes involved in tight junctions and calcium signaling.

### Longitudinal changes associated with remission differentiate non-responders and responders following FMT
The ability of the 78 transcripts that were significantly regulated in responders (Tx0Y vs Tx8Y) to cross-sectionally differentiate between responders and non-responders (Tx8N vs Tx8Y) was assessed. Ordination and multivariate analyses utilizing only the counts of the 78 transcripts showed that the Tx8Y group was significantly different to the other three groups (Tx0N, Tx0Y, and Tx8N) (ANOSIM; Fig. 3A), with the majority of the differences between non-responders and responders after therapy (Tx8N vs

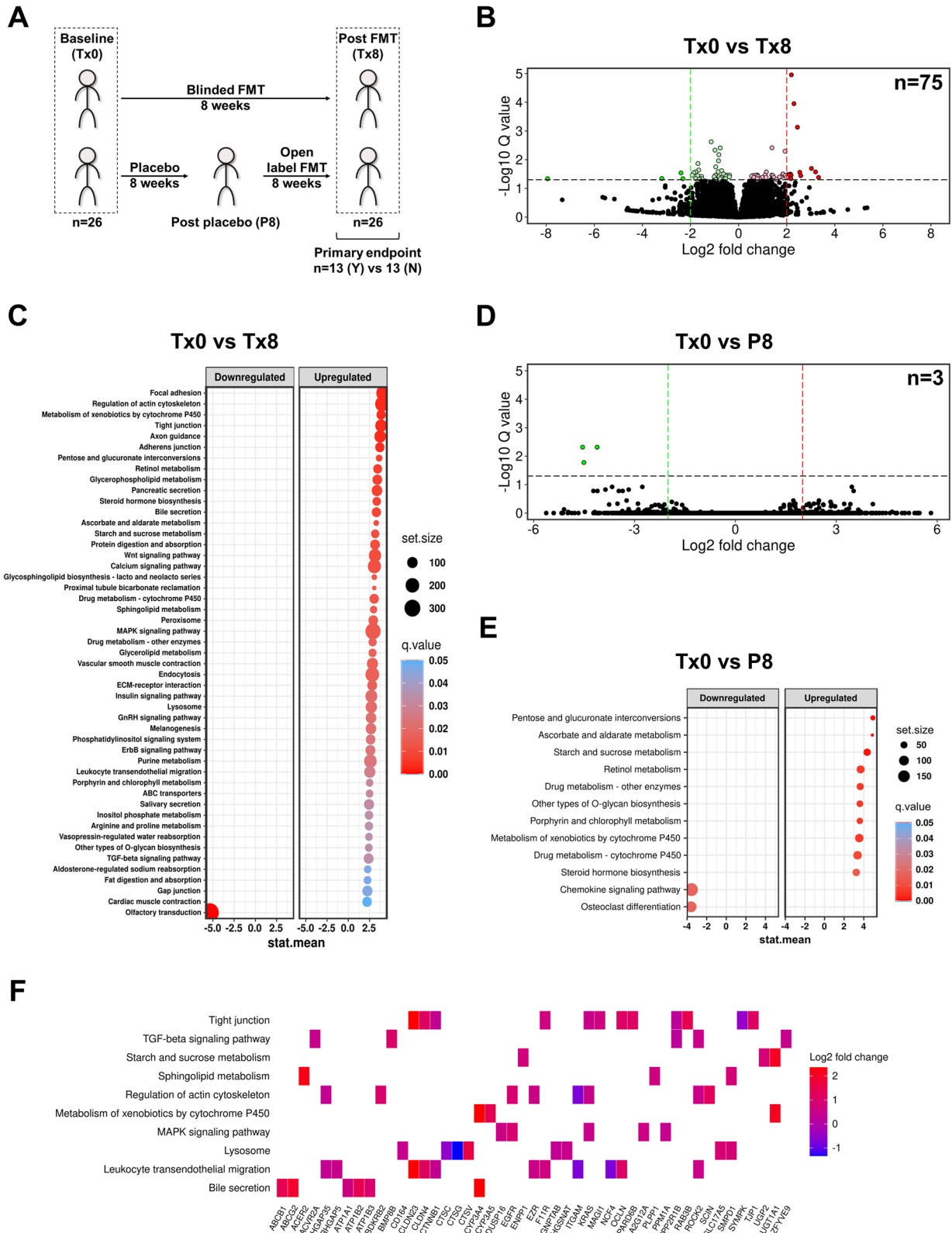

**Fig. 1 | Fecal microbiota transplantation (FMT) has a substantial effect on host colonic mucosal responses. A** Sampling schema of human cohort. Tx0, baseline; Tx8, post FMT; P8, post placebo; Y, yes; N, no. **B** Volcano plot of differential gene expression following treatment with FMT for 8 weeks. Total genes identified by DESeq2 to be significantly regulated $n = 75$ ($q < 0.05$). Red, significantly upregulated; Green, significantly downregulated. **C** Significantly regulated pathways ($q < 0.05$) following FMT treatment for 8 weeks. Regulated pathways were identified using GAGE. **D** Volcano plot of differential gene expression following treatment with placebo for 8 weeks. Total genes identified by DESeq2 to be significantly regulated $n = 3$ ($q < 0.05$). **E** Significantly regulated pathways ($q < 0.05$) following placebo treatment for 8 weeks. **F** Genes identified to be differentially expressed ($q < 0.2$) within significantly regulated pathways of interest.

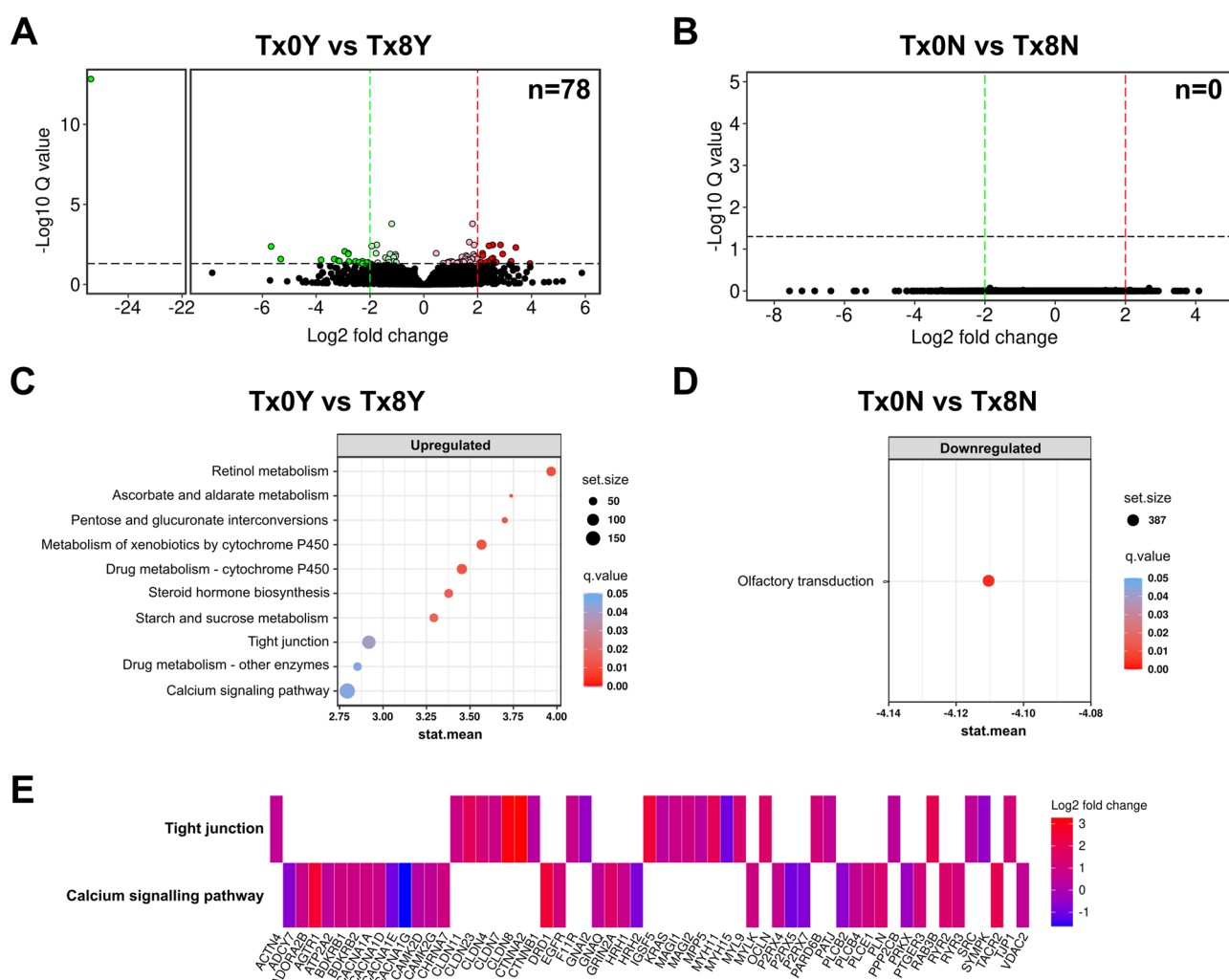

**Fig. 2 | Responders to fecal microbiota transplantation (FMT) have pronounced effect on differential gene expression in their colonic mucosa that is not present in non-responders. A** Volcano plot of differential gene expression in responders following treatment with FMT for 8 weeks. Total genes identified by DESeq2 to be significantly regulated $n = 78$ ($q < 0.05$). Tx0Y, responder baseline; Tx8Y, responder post-FMT. Red, significantly upregulated; Green, significantly downregulated. **B** Volcano plot of differential gene expression in non-responders following treatment with FMT for 8 weeks. Total genes identified by DESeq2 to be significantly regulated $n = 0$ ($q < 0.05$). Tx0N, non-responder baseline; Tx8N, non-responder post-FMT. **C** Significantly regulated pathways ($q < 0.05$) in responders following FMT treatment for 8 weeks. Regulated pathways were identified using GAGE. **D** Significantly regulated pathways ($q < 0.05$) in non-responders following FMT treatment for 8 weeks as identified by GAGE. **E** Top regulated genes ($n = 60$) in pathways comprising the largest number of genes (tight junctions; calcium signaling). Genes were classified by $q$ value and selection corresponded to $q < 0.5$.

Tx8Y) observed on the PCO2 axis (Fig. 3A). To establish how the 78 genes classified patient samples, canonical analysis of principal coordinates (CAP; constrained ordination) was applied which showed that the responders following FMT (Tx8Y) were the only group classified together with high accuracy (Fig. 3B). Of note, the misclassified responder (Fig. 3B table) was a patient who experienced endoscopic response but did not achieve the strict endoscopic remission endpoint by week 8, while the non-responder seen to cluster with responders (Fig. 3A, right) was a patient who reached the clinical remission endpoint but not the endoscopic endpoint (see ref. 8. for endpoint definitions). This suggested that the mucosal molecular response, to some extent, may reflect the gradient nature of the response to therapy. These results were further validated by calculating a resemblance matrix based on Euclidean distances and performing permutational multivariate ANOVA (Pseudo-F = 2.76, $p = 0.0046$, df = 3.48; all PERMDISP $p = 0.34$-$0.99$). Pairwise comparisons showed that the responders following FMT (Tx8Y) were significantly or borderline different to the remaining groups (Tx0Y: $t = 2.24$, $p = 0.0083$; Tx0N: $t = 1.92$, $p = 0.014$; Tx8N: $t = 1.46$, $p = 0.073$).

Given that the differences were observed on the PCO2 axis, which contributed 13.7% of the total variation observed in the data, genes that contributed the most to this axis were assessed. The top three genes that were significantly positively correlated to the axis coordinates were identified to be interferon regulatory factor *IRF4*, ENSG00000287626 (antisense transcript to *DTNBP1*), and guanylate-binding protein *GBP5* (Fig. 3C). Of the three, only *GBP5* showed no sign of regulation whatsoever in non-responders (*IRF4*: log₂ Fold Change = −0.70, $p = 0.061$, $q > 0.99$; ENSG00000287626: log₂ Fold Change = −0.56, $p = 0.29$, $q > 0.99$; *GBP5*: log₂ Fold Change=0.043, $p = 0.92$, $q > 0.99$). Further, *GBP5* and *IRF4* did not show biologically relevant differences between responders and non-responders at baseline (*IRF4*: log₂ Fold Change = 0.013, $p = 0.97$, $q > 0.99$; ENSG00000287626: log₂ Fold Change = 1.45, $p = 0.0041$, $q > 0.99$; *GBP5*: log₂ Fold Change = 0.42, $p = 0.17$, $q > 0.99$), collectively indicating *GBP5* to be a potent discriminatory factor between responders and non-responders.

The findings indicate good concordance between longitudinal and cross-sectional analyses, and that *GBP5* to be a relevant host biomarker of disease remission.

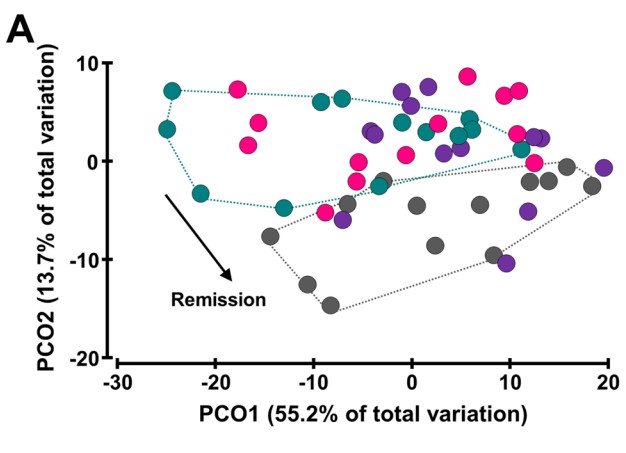

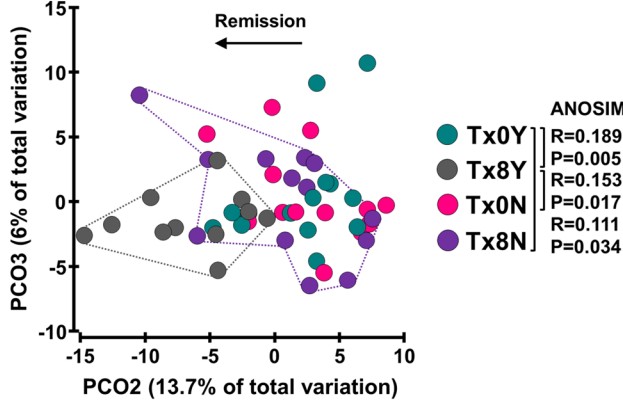

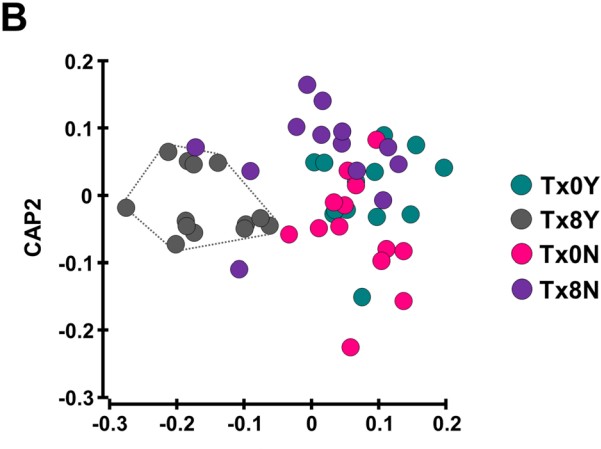

Canonical analysis of principal coordinates

| Group | Correct | Misclassified | Correct (%) |
| --- | --- | --- | --- |
| Tx0Y | 6 | 7 | 46.2 |
| Tx8Y | 12 | 1 | 92.3 |
| Tx0N | 4 | 9 | 38.5 |
| Tx8N | 4 | 9 | 38.5 |

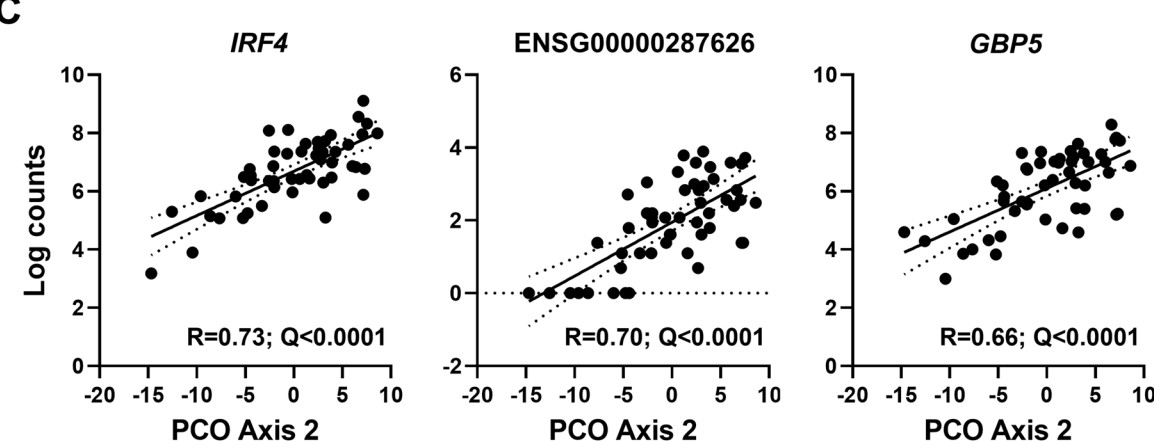

**Fig. 3 | Molecular mucosal changes in responders following fecal microbiota transplantation (FMT) differentiate them from non-responders cross-sectionally.** **A** Principal coordinate analysis (PCO) of differentially expressed genes in responders ($n = 78$ genes). Bray-Curtis similarities were calculated on $\log(x + 1)$ transformed normalized counts of host genes. Inter-group differences were tested using One-Way ANOSIM, with one-tailed significance computed by permutation, and only pair-wise differences with Tx8Y are shown. Tx0N, non-responder baseline (pink); Tx0Y, responder baseline (light blue); Tx8N, non-responder post-FMT (purple); Tx8Y, responder post-FMT (gray). **B** Constrained ordination using Canonical Analysis of Principal coordinates (CAP). Classification of samples according to group is presented in the table. **C** Top three genes significantly ($q < 0.05$) correlated with PCO axis 2. PCO axis 2 was chosen because the majority of variation between responders and non-responders post-FMT was observed on that axis. Pearson correlations were performed, and $p$ values were corrected for false discovery rate ($q$ value) using the Benjamini–Hochberg method. Source data are provided as a Source Data file.

## *GBP5* expression is correlated to bacterial taxon associated with lack of efficacy

Since guanylate-binding proteins function as antimicrobial proteins, modulating host immune pathways in response to stimuli of microbial origin[28], correlations between levels of *GBP5* and the mucosal microbiota were assessed. While no relationship between *GBP5* expression and beta diversity was identified (Pseudo-F = 1.14, $p = 0.19$, df = 58),

*Sutterella* OTU29 was found to be the only taxon to share a significant linear correlation with *GBP5* expression after multiple comparison correction (Supplementary Fig. 5A; Pearson $r = 0.51$, $q = 0.0037$). Further, this taxon was also correlated with *IRF4* expression (Supplementary Fig. 5B). This is particularly relevant given that fecal levels of *Sutterella wadsworthensis* were associated with lack of FMT efficacy in patients and with less effective donor batches in this clinical cohort[25].

## Deletion of *GBP5* results in resistance to chemically induced colitis

To assess the role of GBP5 in colitis, littermate WT and *Gbp5⁻/⁻* mice were bred and cohoused until 24 h prior to exposure to DSS. Mice were exposed to DSS in drinking water for 6 days followed by 4 days of regular drinking water (Fig. 4A). *Gbp5⁻/⁻* mice were more resistant to chemically induced colitis, showing a significantly higher body weight relative to littermate WT controls from day 8 following treatment (Fig. 4B), with a significantly higher colon length at day 10 (Fig. 4C, D). Histological analysis revealed that *Gbp5⁻/⁻* mice had lower histology scores as assessed by inflammation, hyperplasia, ulceration, and extent of tissue damage (Fig. 4E,F), however, this did not reach statistical significance ($p = 0.11$). When stratified by location, all regions showed lower histology scores in *Gbp5⁻/⁻* mice when compared to WT mice (two-way ANOVA: genotype: $p = 0.059$ and colonic region: $p < 0.0001$),

with the middle region reaching statistical significance (Šídák's multiple comparisons test; $p = 0.041$) (Fig. 4G). These differences were not due to variations in the gut microbiota as both genotypes had similar bacterial and fungal profiles at baseline and following DSS (Supplementary Fig. 6A-D). Moreover, while *Sutterella* may play a synergistic role when present within the microbiota as suggested by the human data, taxa classified to *Sutterella* were not detected in any of the mice, and thus, were not essential for the differential colitis phenotype in this model.

Colonic tissue lysates from *Gbp5⁻/⁻* mice were confirmed to lack GBP5 (Fig. 5A). Given that GBP5 and IRF4 were both markers of response in humans, had correlated expression profiles in the patients (Pearson $r = 0.84$, $p < 0.0001$), the levels of phosphorylated and total IRF4 were assessed in the colonic tissue of WT and *Gbp5⁻/⁻* mice at day 10. A significantly higher ratio of phospho-IRF4 to total IRF4 was

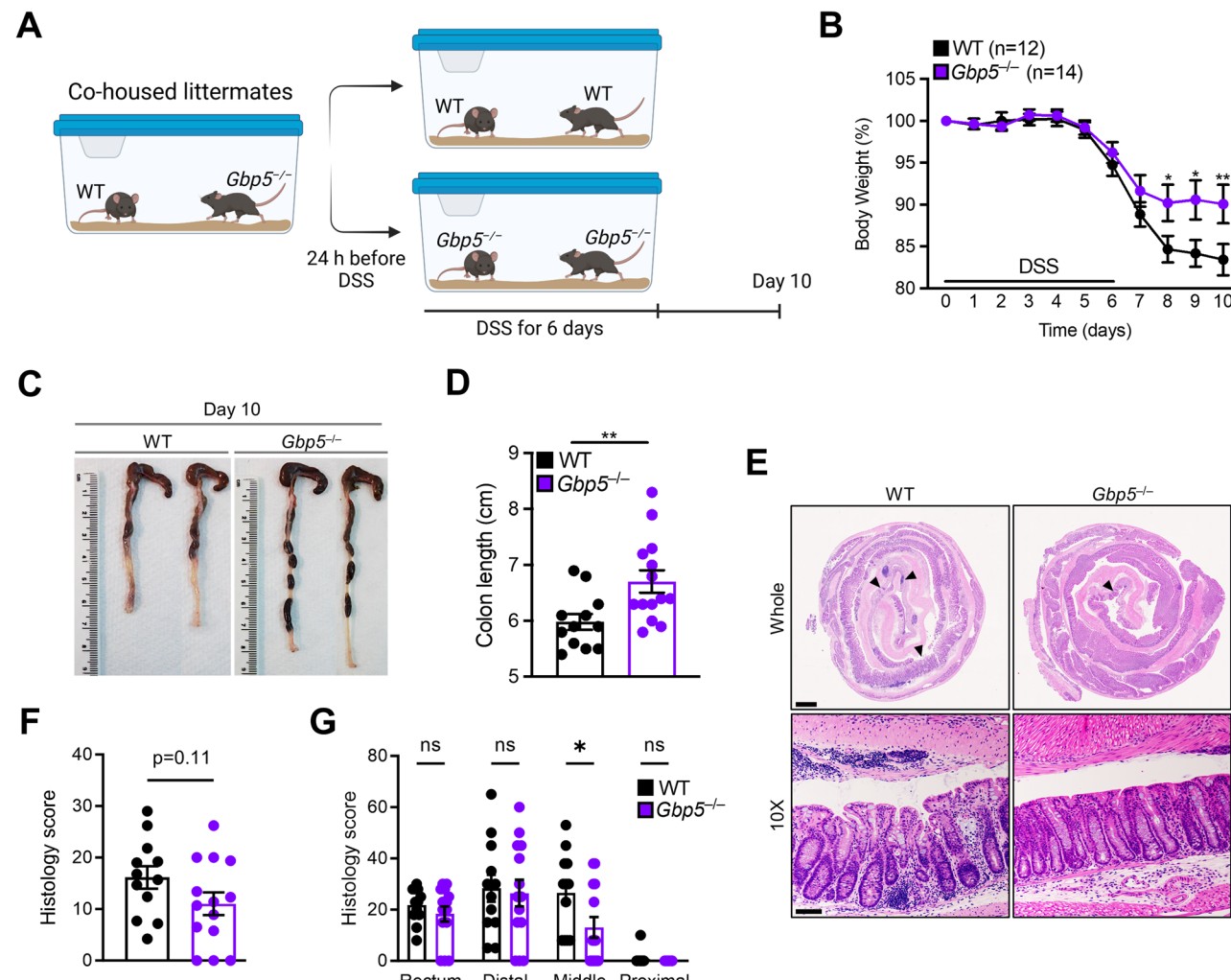

**Fig. 4 | *Gbp5⁻/⁻* mice are more resistant to colitis than wild-type littermates.**
**A** Schema of experimental colitis model using littermate wild-type (WT) and *Gbp5⁻/⁻* mice. Mice were cohoused until 24 h prior to exposure to dextran sulfate sodium (DSS). **B** Change in body weight of mice throughout the 10 days of the model. Differences between groups were tested using Two-Way ANOVA with Šídák's multiple comparisons test. Days 0–7 $p > 0.8$; Day 8: $p = 0.044$; Day 9: $p = 0.0103$; Day 10: $p = 0.0065$. Data are presented as mean ± SEM. *$p < 0.05$; **$p < 0.01$. **C** Representative images of colon lengths for littermate WT and *Gbp5⁻/⁻* mice. **D** Differences in colon lengths between littermate WT (n = 12) and *Gbp5⁻/⁻* (n = 14) mice at day 10. Data distribution was assessed using the Shapiro-Wilk test and differences were tested using a two-tailed unpaired $t$ test ($t = 2.86$, $p = 0.0087$). Data are presented as mean ± SEM. **$p < 0.01$ **E** Haematoxylin and eosin staining of the colonic tissue of mice on day 10. Data are representative of three independent

experiments. Scale bars, 1000 µm (top), 100 µm (bottom). Arrowheads indicate histological damage including inflammation, ulceration and hyperplasia.
**F** Histological score of the colonic tissue of WT (n = 12) and *Gbp5⁻/⁻* (n = 14) mice on day 10 presented as mean ± SEM. Each symbol represents an individual mouse. Data distribution was assessed using the Shapiro-Wilk test and differences were tested using a two-tailed unpaired $t$ test ($t = 1.64$, $p = 0.11$). **G** Histological scores of the colonic tissue of WT (n = 12) and *Gbp5⁻/⁻* (n = 14) mice on day 10 stratified by region and presented as mean ± SEM. Each symbol represents an individual mouse. Differences were tested using a two-way ANOVA with Šídák's multiple comparisons test and statistical significance in the middle region was validated with a two-sided Mann-Whitney test ($U = 41.5$, $p = 0.024$). ns, not significant; *$p < 0.05$. Source data are provided as a Source Data file.

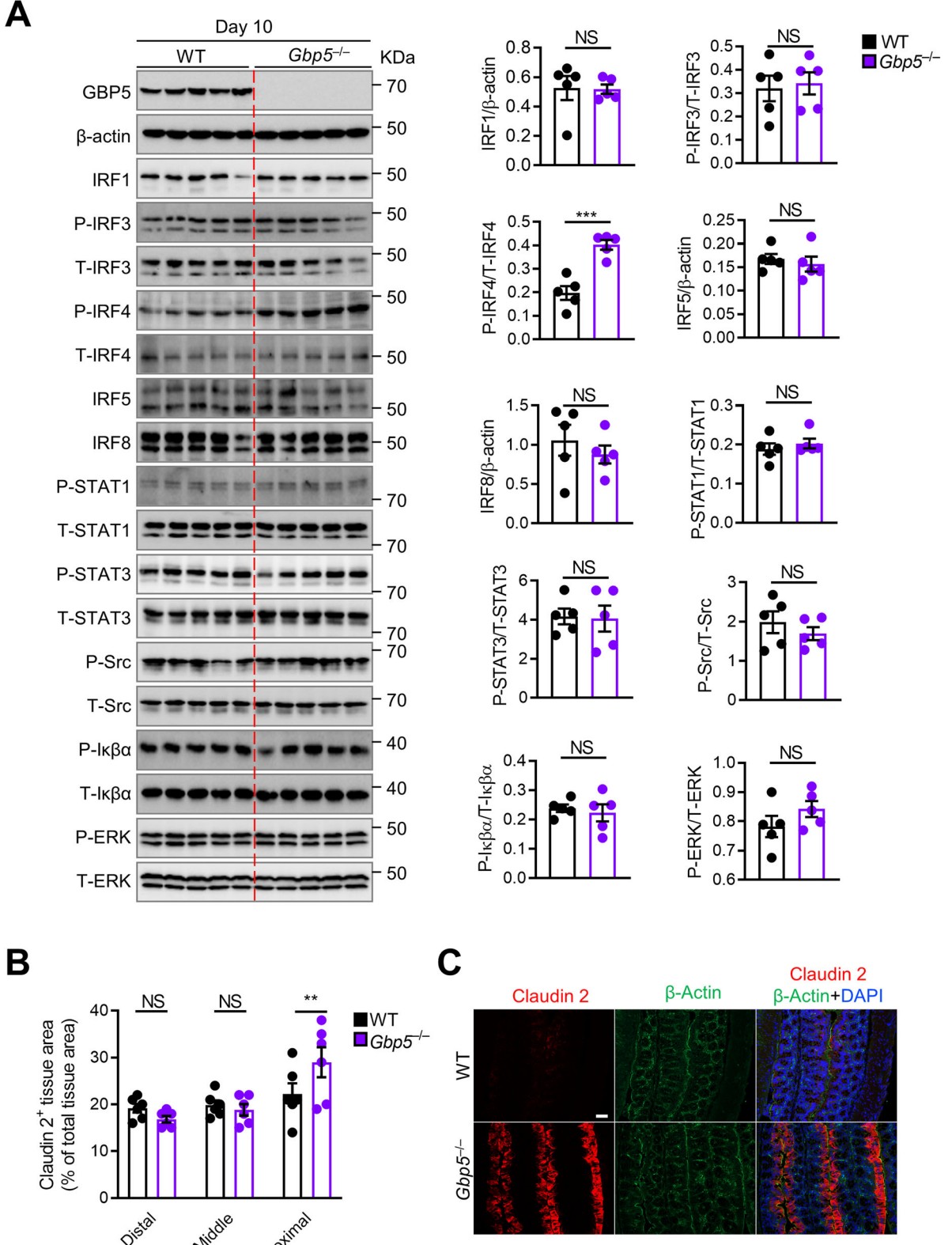

**Fig. 5 | Differential protein regulation between *Gbp5*^−/− and WT littermates.**
**A** Expression and phosphorylation of proteins of interest in littermate WT ($n = 5$) and *Gbp5*^−/− ($n = 5$) mice at day 10. Band volume was employed for densitometry and differences were tested using two-tailed unpaired t-tests following assessment of data distribution by the Shapiro-Wilk test. All $p > 0.22$ except P-IRF4/T-IRF4 $p = 0.0004$. Data are presented as mean ± SEM. NS, not significant; ***$p < 0.001$. **B** Quantification of claudin 2-positive area over total colon tissue area in littermate

WT ($n = 6$) and *Gbp5*^−/− ($n = 6$) mice at day 10. Each symbol represents an individual mouse. NS, not statistically significant; **$p < 0.01$ ($p = 0.0018$) by two-way ANOVA with Šídák's multiple comparisons test following confirmation of data distribution by the Shapiro-Wilk test. Data are presented as mean ± SEM.
**C** Immunohistochemical staining of claudin 2, β-Actin and 4′,6-diamidino-2-phenylindole (DAPI) in the colon tissue of littermate WT and *Gbp5*^−/− mice on day 10. Scale bar, 50 μm. Source data are provided as a Source Data file.

identified in $Gbp5^{-/-}$ mice when compared to WT mice (Fig. 5A, Supplementary Fig. 7). No differences between $Gbp5^{-/-}$ and WT mice were observed for other related kinases and signaling molecules (Fig. 5A, Supplementary Fig. 7). The results identify a role for GBP5, and potentially IRF4, in the establishment of colitis in mice. This is in line with our findings that downregulation of *GBP5* and *IRF4* was associated with remission of UC following FMT in humans.

### Resistance to colitis in $Gbp5^{-/-}$ mice is associated with tight junction regulation

Next, differences in $Gbp5^{-/-}$ mice that may be mechanistically associated with increased resistance to colitis were explored. The tight junction marker claudin 2 was examined in areas where inflammation, hyperplasia and ulceration were detected on histological assessment (i.e., middle and distal colon). Tight junctions were a key pathway identified to be regulated in disease remission (Fig. 2C), with a number of claudins observed to be upregulated (Fig. 2E), and claudin 2 has established mechanistic links to colitis progression. No difference in claudin 2 expression was found in either area (Fig. 5B). We speculated that any molecular variations may precede disease onset and tissue damage may obscure detectable differences. Thus, differences in claudin 2 were assessed in the proximal colon where no tissue damage was detected in either group, and a significant difference between $Gbp5^{-/-}$ and WT mice was observed (Fig. 5B,C). To further assess this relationship, we measured levels of claudin 2 expression in WT mice prior to (day 0) and following exposure to DSS (day 10). We identified significant decreases in claudin 2 levels in the distal and middle colon of mice following development of colitis (Supplementary Fig. 7B). This decrease was not significant ($p = 0.065$) in the proximal colon where no signs of colitis were detected (Supplementary Fig. 7B), indicating that claudin 2 levels were associated with histological detection of colitis, and that they may be key to colitis resistance in $Gbp5^{-/-}$ mice.

## Discussion

The host colonic mucosal transcriptome of patients with UC prior to and following FMT was examined to identify molecules that are impacted by FMT and those associated with disease remission. FMT had a robust effect on the mucosal transcriptome of patients with UC, and this was not observed in patients receiving placebo. Consistent differential gene expression following FMT was identified in patients who responded to therapy but not in non-responders, with pathways such as tight junctions and calcium signaling found to be regulated. In contrast, alternative splicing events were observed in both responders and non-responders to FMT, albeit some events such as those within *SYK* were differentially present in these subgroups. Assessment of the capacity for longitudinal changes associated with response to inform on cross-sectional differences between responders and non-responders revealed that the expression of *GBP5* and *IRF4* to be strongly associated with remission. Genetic deletion of *GBP5* in mice showed that mice without GBP5 were more resistant to developing colitis than wild-type littermates, with concomitant regulation of IRF4 phosphorylation and the tight junction marker claudin 2 in their colonic tissue.

While FMT is less effective at treating UC than *C. difficile* infection, with success rates ranging from 30–50% rather than ~90%[12–16], FMT, unlike placebo, induced a robust and consistent mucosal response in patients with UC. 49 pathways were significantly regulated by FMT, the majority ($n = 48$) being upregulated, of which structural pathways related to the cytoskeleton and tight junctions were among the most significant. Altered tight junctions and impaired barrier function are well known characteristics of UC often resulting from dysregulated inflammatory signals[29,30] but can also lead to increased luminal antigen uptake that can then promote inflammation[31]. Commensal members of the microbiota and their metabolites (e.g., short chain fatty acids and secondary/tertiary bile acids) can improve barrier integrity through

modulation of tight junction proteins[32]. Notably, on stratification of patients according to clinical outcome, the effect of FMT on tight junctions was only present in responders, while non-responders did not appear to mount any consistent mucosal gene expression profile to FMT. This would suggest that while microbial markers associated with response have been identified[19,25], the colonic mucosal expression profile may be a more effective discriminator of clinical outcome.

In contrast to gene expression, patients from both clinical outcomes showed evidence of alternative splicing events. However, events within responders appeared more consistent as they could be defined into significantly enriched pathways. Spleen-associated tyrosine kinase *SYK* was of particular interest as the exon skipping event found to be significantly present in responders and not in non-responders is predicted to lead to differential levels of the long and short forms of the protein, with these isoforms having a different capacity to regulate cell survival[33]. SYK is not only involved in adaptive immunity but is also key to innate immune recognition of microbial species and downstream signaling[27]. Further, inhibition of SYK with fostamatinib attenuates inflammation and disease activity in a rat model of acetic acid-induced colitis[34].

The seventy-eight genes regulated by FMT in responders were found to delineate clinical outcome both longitudinally and cross-sectionally, and through this analysis, the expression of *GBP5* and *IRF4* were found to be among the most discriminating variables explaining the variation across groups. GBP5 belongs to the family of guanylate-binding proteins capable of sensing and lysing microbes[35]. It has been shown to be important in the pathogenesis of both bacterial and viral infections[36–40], and recently found to be upregulated in patients with UC[41]. Specifically, *GBP5* was upregulated 8.1-fold in colonic biopsies from European patients with UC when compared to healthy controls and upregulated 3.8-fold in colonic biopsies from Chinese patients with UC when compared to healthy controls[41]. Intriguingly, *IRF4* expression has also been previously shown to be upregulated in patients with UC and to play a role in experimental colitis in mice[42,43]. In support, we found that genetic deletion of *GBP5* protected mice from colitis when they were compared to their wild-type littermates, and additionally IRF4 phosphorylation was dysregulated in their colonic tissue. We speculate that in responders, FMT displaced the microbial trigger inducing *GBP5* expression and downstream inflammatory responses. The downregulation of *GBP5* subsequently decreases colonic inflammation in these patients. In mice, DSS is known to injure the colonic mucosal barrier due to its toxicity to epithelial cells, leading to increased immune cell exposure to microbial antigens. In the absence of GBP5, downstream immune cell activation and inflammation is diminished, resulting in decreased pathology. IRF4 appears to be tightly linked to GBP5-associated inflammation.

In support, we observed that the tight junction marker claudin 2 was significantly higher in undamaged colonic tissue of GBP5-deficient mice when compared to WT mice. While higher expression of claudin 2 has been reported in several contexts, such as being involved in progression of immune-mediated colitis[44], increased claudin 2 expression through IL-22 reduces the severity of colitis induced by infection through pathogen clearance[45]. More relevant, while colonic claudin 2 expression increased permeability, it also increased colon length and protected against DSS colitis through lower colonocyte death on colitis, lower immune activation, and increases in regulatory T cells[46]. This is in line with our findings in humans where the tight junction pathway was significantly enriched in data from patients who achieved remission.

This study has several limitations. The number of patients included in the transcriptome analysis may have influenced the power to detect gene expression changes associated with FMT and with clinical outcomes. To account for the lower number of responders available from the clinical trial, a 1:1 matched analysis between responders and non-responders was conducted. Moreover, lowering of the statistical

threshold to $q < 0.1$ did not alter the conclusions. The transcriptome data was generated using bulk RNA sequencing, and while cell type composition was inferred to be similar across comparisons, an effect of cell type composition on the results of gene expression cannot be discounted. The mouse model within this study examined the role of the molecules of interest in the development of colitis and not the response to FMT. Given that the loss-of-function model resulted in resistance to colitis, a mouse model whereby susceptibility to colitis is rescued by FMT was not possible. Our collective findings would suggest that the concerted role of GBP5 and IRF4 extends beyond the response to FMT to a larger role in the development of colitis; however, further studies are required to confirm this.

In conclusion, there is a consistent colonic mucosal response following FMT encompassing structural pathways such as tight junctions with the majority of the measured response being attributed to patients with a positive clinical outcome. The GBP5 appears to be a key modulator of colitis in humans and mice.

## Methods

### Study cohort
Patients with UC were previously enrolled in a double-blind, randomized, placebo-controlled trial to test the efficacy of 8-week intensive FMT therapy, with those receiving placebo then treated with open-label FMT for 8 weeks[8]. This investigator-initiated study was sponsored by the University of New South Wales (UNSW), approved by the St Vincent's Hospital Sydney Human Research Ethics Committee (HREC/13/SVH/69) and registered with ClinicalTrials.gov (NCT01896635) and the Australian Therapeutic Goods Administration Clinical Trial Notification Scheme (2013/0523). Written informed consent was obtained from all participants.

Gut mucosal samples were collected from patients at baseline (Tx0), at the end of placebo if applicable (P8), and patients at the end of FMT (Tx8). Baseline mucosal biopsies were taken from the most inflamed segment of the rectosigmoid. Longitudinal sampling of patients following FMT or placebo was standardized according to baseline biopsies. A subset of the cohort consisting of 60 samples were matched for response ($n = 13$ vs $13$) and selected for analysis of host gut mucosal response to therapy. Criteria for the patient and sample selection are outlined in Supplementary Fig. 8. We aimed to analyze the highest possible number of responders to FMT, matched 1:1 with non-responders according to baseline disease activity. Of the 26 patients ($n = 12$ female) selected for analysis, mean age ± SEM = 40.8 ± 2.3 years (95% CI = 36.0–45.6). $n = 18$ (9 female) and $n = 8$ (3 female) patients were in the blinded (directly on FMT) and open-label (i.e., initially on placebo) arms, respectively.

### RNA extraction from mucosal biopsies and shotgun sequencing
Colonic biopsy samples were homogenized and nucleic acids extracted using the Macherey-Nagel Nucleospin RNA Isolation Kit (Catalog no. 740955) as previously described[25]. Colonic RNA was then purified from DNA using the MOBIO On-Spin Column DNase kit (Catalog no. 15100) and Macherey-Nagel RNA clean-up kit (Catalog no. 740948)[25]. RNA was prepared for sequencing using the TruSeq Stranded Total RNA-seq Ribo-zero Gold (Catalog no. RS-122-2301) sample preparation kit and sequenced using Illumina NovaSeq 6000 S2 2×100 bp chemistry at the Ramaciotti Centre for Genomics (UNSW Sydney).

### Raw data and statistical analysis
RNA sequencing data was received using proprietary Illumina software and stored locally. Raw reads as fastq files were quality checked using fastqc v0.11.5 (RRID:SCR_014583). Salmon v1.2.1[47] and the human GRch38 reference were used to quantify the expression of transcripts within the data and SeqMonk v1.47.1 (RRID:SCR_001913) was employed to assess the mapped data. Salmon options included automatic detection of library type, selective alignment (validateMappings) and

correction for GC bias. Cell composition was inferred from normalized count data using CIBERSORTx[48]. Bray-Curtis dissimilarities were calculated, and the resemblance matrix was ordinated using principal coordinate analysis (PCoA). Differences across groups were determined using Analysis of Similarities (ANOSIM) on Primer-e v6. DESeq2 v1.30.1[49] was utilized to identify significantly differentially expressed genes across comparisons using both unpaired and paired approaches. GAGE v2.40.2[50] was then used for pathway analysis of the differentially expressed genes, limiting the analysis to coding genes (i.e., excluded non-coding transcripts). A benefit of employing GAGE is that it does not limit the analysis to genes that are significantly differentially expressed and instead assesses the fold-change across the whole transcriptome. rMATs v4.0.2[51] was employed to identify differences in splicing events across groups. Default parameters were used (paired; readLength 100; libtype fr-firststrand) and dependences included python v2.7.15, gsl v1.16, star v2.7.2b and samtools v1.10. Pathway analysis of genes with splicing events was performed using Enrichr[52] with KEGG 2021 Human pathways as the reference database. Plots (volcano, pathways, heatmaps, splicing event and protein structure) were generated using the R packages ggplot2 v3.3.3 (Wickham H, ISBN: 978-3-319-24277-4), maser v1.8.0 (F.T. Veiga D; DOI: 10.18129/B9.bioc.maser), and drawProteins v1.10.0[53].

For assessment of concordance between longitudinal and cross-sectional expression profiles, counts were limited to genes found to be significant, $\log(x + 1)$ transformed, and Bray–Curtis dissimilarities calculated. The resemblance matrix was visualized using PCoA and differences across groups determined using ANOSIM. Canonical analysis of principal coordinates (CAP) was performed in Primer-e v6 to classify our observations against the Treatment x Response variable (Tx0N, Tx0Y, Tx8N, and Tx8Y) and obtain the level of misclassification error within each group when only significantly regulated genes were considered. The results were validated by calculating Euclidean distances between samples and testing differences using permutational multivariate ANOVA (PERMANOVA). Dispersion effects within the PERMANOVA were ruled out by testing homogeneity of multivariate dispersions using PERMDISP. To determine which genes were correlated with the principal coordinate axis, Pearson correlations were calculated in GraphPad Prism v9 and P-values corrected for false discovery rate using the Benjamini-Hochberg method.

### Human mucosal microbiota profiling and analysis
16S rRNA gene amplicon sequencing (V1-3 region) of DNA extracted from the mucosal biopsies was performed previously on an Illumina MiSeq platform (2 × 300 bp chemistry)[25]. DNA was extracted at the same time RNA was extracted above. While fecal shotgun metagenomic data are available for these patients, we analyzed mucosal 16S rRNA gene amplicon sequencing data as they were derived from the same nucleic acid extract as the RNA sequencing data. Raw 16S rRNA gene reads were analyzed using Mothur v1.44.2[54] and vsearch v2.13.3, employing the MiSeq standard operating procedures with some minor modifications[55]. Taxonomic reference used was RDP v18. Final read depth for 16 S rRNA gene data was 12274 clean reads/sample. Alpha diversity (species richness, species evenness and Shannon's diversity index) was calculated using Primer-E v6 and differences between groups tested using GraphPad Prism v9. Beta diversity was interrogated by calculating Bray-Curtis similarities from OTU relative abundances (%) that were transformed by square-root. PCoA and Analysis of Similarities (ANOSIM; One-Way or Two-Way with 9999 permutations) were then calculated and visualized using Primer-E v6. To examine the correlations between the microbiota and GBP5, Pearson correlations between the top 100 bacterial OTUs and normalized expression counts of *GBP5* and *IRF4* were calculated in GraphPad Prism v9 and P-values corrected for false discovery rate.

### Gbp5$^{-/-}$ mice and experimental colitis model

Mice with a genomic deletion of GBP5 (called Gbp5$^{-/-}$) were generated on the C57BL/6NcrlAnu background by Cas9/CRISPR-mediated genome editing technology and characterized in a previous study[28]. Briefly, Cas9 protein (Cat#:1081059) and single guide RNAs (sgRNA) were purchased from IDT (Singapore). The following sequences Gbp5 sgRNA1 5′- ATTGTGGGTCTTTATCGCAC AGG-3′, Gbp5 sgRNA2 5′- CTCAAACATTCAATCTACCG CGG-3′ and Gbp5 sgRNA3 5′-CTGCCCGGCTCGAAGCACAG AGG-3′ targeted exons 2, 6, and 10, respectively, resulting in a 7268 bp deletion. The nucleases were delivered into the pronucleus of C57BL/6NCrl fertilized zygotes and incubated overnight at 37 °C under 5% $CO_2$ and two-cell stage embryos were surgically transferred into the ampulla of the pseudo-pregnant CFW/Crl mice. A 7268 bp deletion between exon 2 and 10 of Gbp5 was confirmed using Sanger sequencing at the Biomolecular Resource facilities at the Australian National University.

Mice were bred and maintained at The Australian National University under specific pathogen-free conditions. Female littermate wild-type (WT) ($n = 12$) and Gbp5$^{-/-}$ ($n = 14$) mice of 8–10 weeks of age were given 1.5% dextran sulfate sodium (DSS) (MP Biomedicals, #160110) in their drinking water for 6 days, followed by regular drinking water for 4 days. Mice were ethically culled on day 10 by $CO_2$ inhalation or cervical dislocation. The colon was harvested from all mice and colon length was measured. All animal studies were conducted in accordance with the Protocol Number A2020/18 approved by The Australian National University Animal Experimentation Ethics Committee.

### Immunoblotting analysis of mouse colon tissue

Colon tissues were homogenized in 1 ml radioimmunoprecipitation assay buffer[56] using an Omni TH-2 tissue homogenizer, and lysates were cleared by centrifugation at $12,000 \times g$ for 10 min at 4 °C. Protein concentrations were normalized using Pierce™ BCA Protein Assay Kit (ThermoFisher Scientific, #23227). 15-20 μg of protein in 4× sample loading dye were separated on 8–12% polyacrylamide gels using the Trans-Blot Turbo system (Bio-Rad), followed by transfer onto the polyvinyldifluoride membranes (Millipore, # IPVH00010).

Primary antibodies used for immunoblotting were β-actin (Cell Signalling Technology, #4970), GBP5 (1:1000)[57], Phospho-Src (Tyr527) (New England Biolabs, #2105S), Src (New England Biolabs, #2108S), Phospho-ERK1/2 Thr202/Tyr204 (Cell Signalling Technology, #9101), ERK1/2 (Cell Signalling Technology, #9102), Phospho-IkB (Cell Signaling Technology, #2859), IkB (Cell Signaling Technology, #9242), IRF1 (Cell Signaling Technology, #8478), Phospho-IRF3 (Cell Signaling Technology, #4947), IRF3 (Cell Signaling Technology, #4302), Phospho-IRF4 Tyr122/Tyr125 (ThermoFisher Scientific, #PA5105214), IRF4 (New England Biolabs, #4948S), IRF5 (Abcam, #ab33478), IRF8 (Cell Signaling Technology, #5628), Phospho-STAT1 (Cell Signaling Technology, #9167), STAT1 (Cell Signaling Technology, #14994), Phospho-STAT3 (Cell Signaling Technology, #9145), and STAT3 (Cell Signaling Technology, #9139). Secondary antibodies used for immunoblotting were Peroxidase AffiniPure (Jackson ImmunoResearch, #111-035-045) and Peroxidase AffiniPure (Jackson ImmunoResearch, #115-035-146). Protein bands were visualized with Clarity Western ECL Substrate (Bio-Rad, #1705061) or SuperSignal™ West Atto Ultimate Sensitivity Substrate (ThermoFisher Scientific, #A38554) on a Chemi-Doc™ Imaging System (Bio-Rad). Image processing and densitometric quantification of protein bands were performed using the Image Lab software (Bio-Rad). Both lane and band volume analyses were presented.

### Histology analysis

Mouse colons were rolled into a "swiss roll" and fixed in 10% neutral buffered formalin, processed, and then embedded in paraffin. Blocks were sectioned at a thickness of ~5 μm onto microscopy slides and stained with haematoxylin and eosin. Histology scores were assigned based on the parameters: inflammation, hyperplasia, ulceration, and extent of damage[58]. For example, 60% ulceration = ulceration score of 60. For inflammation and hyperplasia, the qualitative histological parameters were converted into quantitative data as 0 = normal; 25 = mild; 50 = moderate; 75 = marked; 100 = severe. The tissue sections were divided into proximal (40% of tissue area), middle (30% of tissue area), and distal (20% of tissue area) and rectal (10% of tissue area) portions. Histology scores were then calculated separately in all the portions, and as a combined total. For example, mild inflammation in proximal, moderate inflammation in distal, moderate inflammation in middle, and mild inflammation in rectal would translate to an overall inflammation score of 25 ×40% + 50 ×30% + 50 ×20% + 25 ×10% = 37.5.

### Immunohistochemistry

Tissue section slides were deparaffinized in xylene (534056, Sigma-Aldrich) and rehydrated in decreasing concentrations of ethanol (107017, Merck) before being washed in phosphate-buffered saline (PBS, pH 7.1-7.5) [D8537, Sigma-Aldrich] and Tris-buffered saline containing 1% Tween-20 (TBST) [200 mM Tris-HCl, pH 7.4, 1.37 M NaCl, 0.1% (v/v) Tween-20]. Slides were subjected to antigen retrieval by heating in sodium citrate buffer (10 mM sodium citrate, pH 6.0, 0.05% Tween-20) at 95 °C for 10 min followed by cooling at room temperature for 30 min. The slides were washed for 5 min in TBST followed by distilled water for 5 min, then incubated for 1 h in blocking buffer (10% Normal Goat Serum [005000121, Jackson ImmunoResearch] in PBS supplemented with 0.3% Triton-X100). Slides were incubated overnight at 4 °C with primary antibodies diluted in antibody dilution buffer (1% Bovine Serum Albumin [BSA, A7030, Sigma-Aldrich] in PBS supplemented with 0.1% Triton-X100). Primary antibodies employed targeted claudin 2 (1:100; Abcam, ab53032) and β-actin (1:200; Abcam, ab8226). After incubation with primary antibodies, slides were washed with TBST for 5 min thrice and incubated at room temperature for 1 h in antibody dilution buffer containing 4,6-diamidino-2-phenylindole (DAPI) [1 μg/ml; D9542, Sigma-Aldrich] and secondary antibodies. Secondary antibodies used were Rhodamine RedTM-X conjugate (1:300; Jackson ImmunoResearch, 111-295-144) and Alexa Fluor 488 AffiniPure (1:300; Jackson ImmunoResearch, 115-545-003). Slides were washed with TBST for 5 min thrice and once with distilled water for 5 min, covered with a 22 × 22 mm coverslip (0101050, Superior Marienfeld) using ProLong Gold Antifade mounting media (P36930, ThermoFisher Scientific), and then left to dry overnight in the dark. Samples were visualized using the Zeiss Axio Observer microscope. Image analysis and quantification were performed using Fiji (National Institutes of Health, USA). The integrated density of tissue areas containing the fluorescence signal of claudin 2 was calculated and divided by the integrated density of the whole tissue area containing the fluorescence signal of claudin 2, β-actin (control protein) and DAPI. Investigators were blinded from the genotype of the mice during image acquisition and quantification.

### Mouse fecal 16S rRNA gene and ITS region amplicon sequencing and analysis

The V4 region of the 16S rRNA gene and the fungal ITS2 region were profiled using Illumina MiSeq 2 × 250bp chemistry[59]. DNA was extracted from fecal samples using the QIAamp PowerFecal Pro DNA Kit (Qiagen; Catalog no. 51804). The V4 region of the 16S rRNA gene was amplified using the Earth microbiome primers (515F and 806R) and the Kapa HiFi HotStart ReadyMix (95 °C for 3 min; 25 cycles of 95 °C for 30 s, 55 °C for 30 s, and 72 °C for 30 s; followed by a final step of 72 °C for 5 min). The ITS2 region was amplified using the primers fITS7 and ITS4. Indices and Illumina sequencing adapters were attached using the Nextera XT index kit, and sequencing was performed at the Ramaciotti Centre for Genomics. DNA sequencing data was received using proprietary Illumina software and stored on the BaseSpace cloud

platform. Raw 16S rRNA gene reads were analyzed using Mothur v1.44.2[54] and vsearch v2.13.3, employing the MiSeq standard operating procedures with some minor modifications[55]. Raw ITS2 reads were screened by quality, not aligned, clustered at 5% by abundance. Taxonomic references used were RDP v18 for the 16S rRNA gene and UNITE v6 for ITS2. Final read depth for 16S rRNA gene data and ITS2 data were 13385 and 17526 clean reads/sample, respectively. Differences in alpha and beta diversity were examined using Primer-E v6 and GraphPad Prism v9 as above.

### Reporting summary
Further information on research design is available in the Nature Portfolio Reporting Summary linked to this article.

## Data availability
The reused human mucosal microbiota data have been deposited in the European Nucleotide Archive (ENA) database under accession code PRJEB26473. The mouse fecal microbiota data (both 16S rRNA gene and ITS2 region) generated in this study have been deposited in ENA under accession codes PRJEB55997 for 16S rRNA gene and PRJEB72841 for the ITS2 region. They can also be accessed in the Zenodo database under accession code 10128069. The RNA sequencing data generated in this study have been deposited in ENA under accession code PRJEB73377. The processed RNA sequencing data generated in this study have been deposited in the Zenodo database under accession code 10622773. Databases used in this study can be accessed through the following links: the human GRch38 reference genome [https://www.ncbi.nlm.nih.gov/datasets/genome/GCF_000001405.26/], RDP v18 [https://mothur.org/wiki/rdp_reference_files/], SILVA, and UNITE v6 [https://mothur.org/wiki/unite_its_database/]. All other data are provided in the main text, supplementary information, or from the corresponding author upon request. Source data are provided with this paper.

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

## Acknowledgements

The authors would like to acknowledge the patients and donors who took part in the clinical trial, and the hospital staff that assisted with procedures. This study was supported by the Crohn's & Colitis Foundation of America (Litwin Award; IBD-0391R to N.O.K., M.A.K. and T.J.B. and 988415 to N.O.K., N.C.R., and S.P.) and National Health and Medical Research Council of Australia (Ideas grants APP2011047 to N.O.K. and APP2002686 to S.M.M.). S.P. is supported by an NHMRC Investigator Grant. N.C.R. is supported by a Cancer Institute NSW Early Career Fellowship (2019/ECF1082) and a UNSW Scientia Fellowship. S.M.M. is supported by the Australian National University and a CSL Centenary Fellowship. N.O.K. is supported by a UNSW Scientia fellowship. The funders had no role in study design, data collection and analysis, decision to publish, or preparation of the manuscript. Figure 4A was created using Biorender.

## Author contributions

Conceptualization: N.O.K., S.M.M. Data curation: L.D.W.L., A.P., S.P., C.N. Formal Analysis: L.D.W.L., A.P., S.P., N.O.K. Funding acquisition: N.O.K., S.M.M., S.P., M.A.K., T.J.B. Investigation: L.D.W.L., A.P., S.P., C.N., N.C.R., C.L., S.M.M., N.O.K. Supervision: N.O.K., S.M.M., M.A.K., T.J.B. Visualization: L.D.W.L., A.P., N.O.K., S.M.M. Writing—original draft: N.O.K., L.D.W.L., A.P., S.M.M. Writing – review & editing: S.P., C.N., N.C.R., C.L., M.A.K., T.J.B. All authors approved this version of the manuscript.

## Competing interests

S.P. has served as a consultant for Finch Therapeutics and has received speaker fees from Ferring, Janssen and Takeda. T.J.B. has a pecuniary interest in the Centre for Digestive Diseases, is a medical advisor to Finch Therapeutics, RedHill Bio and Topelia Aust, and holds patents in FMT treatment. All other authors have no conflicts of interest to declare.
