## [Peer Review File · Nature Communications]

Profiling the colonic mucosal response to fecal microbiota transplantation identifies a role for GBP5 in colitis in humans and miceEditorial Note: This manuscript has been previously reviewed at another journal that is not operating a transparent peer review scheme. This document only contains reviewer comments and rebuttal letters for versions considered at *Nature Communications*.

REVIEWER COMMENTS

Mediation comments from reviewer #4 assessing whether reviewer #1 comments have been addressed.

I have re read the comments of reviewer 1 specifically.

Comment 1 : The selection of the groups is adequately outlined.

Comment 2: the analytics is reasonably explained.

Comment 3: The mouse section has improved by adding additional detail. However I agree with reviewer 1 that this section is still not strong in terms of mechanism but can be seen as a confirmatory supplement.

The mouse numbers of 12 and 14 mice for body weight and colon lengths are reasonable (Figure 4B&D). In contrast the histology score shows only 6 mice and is not convincing.

The Swiss role overview 4E is not clear - the higher magnification shows differences. In hindsight this histology assessment needs revised and histology for all animal investigated.

Figure 5 is based on only 5-6 mice per group.

Figure 5&6: The statistics is not always clear - is Anova an appropriate test? Normal distribution and variance criteria are not met on a few occasions in figure 4&5.

Reviewer #2 (Remarks to the Author):

The authors responded to the reviewer's comments, and the manuscript is now somewhat improved. However, there remain some flaws.

In the response to comment#2: the authors provided new data; the expression of claudin2 was increased in the absence of Gbp5. But it remains unclear how GBP5, together with IRF4, regulates the expression of claudin2.

In the response comment#5: The authors did not provide any evidence of how GBP5 and IRF4 interact with each other to maintain the gut homeostasis.

Thus, the findings are interesting. However, the mechanistic insight into how the GBP5/IRF4 axis regulates the intestinal homeostasis is still missing.

Reviewer #3 (Remarks to the Author):

The authors have adequately addressed my concerns.

Reviewer #4 (Remarks to the Author):

Luu et al investigate in their manuscript "Profiling the colonic mucosal response to fecal microbiota transplantation identifies a role for GBP5 and IRF4 in colitis in humans and mice" the transcriptomic response to FMT and perform follow up analysis in animal studies.

GBP5 and IRF4 downregulation was associated with remission. Mice with a defect in GBP5 were more resistant to experimental colitis suggesting a causal link between the therapeutic impact of FMT, regulation of GBP5, and intestinal inflammation.

Comments:

The classification of therapeutic success seems overinflated in Figure 3A and B - 92% correct classification looks optimised since the components substantially overlap.

Please further clarify the location of sampling for biopsies and whether was it the same for all samples.

"Baseline mucosal biopsies were taken from the region identified to be most inflamed in all

patients. Longitudinal sampling of patients following FMT or placebo was standardized according to baseline biopsies. "

The authors state "Notably, analysis of the specific cell-types suggested an underpowered difference in the inferred abundances of naïve and memory CD4 T cells (Supplementary Figure 2D),

>an underpowered difference is no difference?

The responders have a dramatically different transcriptomic response > is this difference similar in the placebo response or a mesalazine or anti-TNF response?

Statistics- some figures likely need revision:

Figure 4 D&f statistics - unpaired t-test Figure 5b Anova - those groups do not look normal distributed and t test/Anova are likely not appropriate

-> Mann Whitney might be applied - but the significance level will certainly differ given the small number and significant overlap

We thank the Reviewers for the constructive comments on our manuscript.

Mediation comments from reviewer #4 assessing reviewer #1 comments

I have re read the comments of reviewer 1 specifically.

Comment 1 : The selection of the groups is adequately outlined.

Comment 2: the analytics is reasonably explained.

Comment 3: The mouse section has improved by adding additional detail. However I agree with reviewer 1 that this section is still not strong in terms of mechanism but can be seen as a confirmatory supplement.

The mouse numbers of 12 and 14 mice for body weight and colon lengths are reasonable (Figure 4B&D). In contrast the histology score shows only 6 mice and is not convincing.

The Swiss role overview 4E is not clear - the higher magnification shows differences. In hindsight this histology assessment needs revised and histology for all animal investigated.

We have now performed histological analysis on all 26 mice within this study. Please see revised Figure 4 panels F & G for the full analysis. We have also modified the manuscript text and figure legend to reflect these revisions. The appropriate statistical analyses have been performed and all details in support have been included in the source data file.

We have also replaced the current Swiss roll figures in Figure 4E with higher quality versions in the revised figure.

Figure 5 is based on only 5-6 mice per group.

Yes, we randomly selected representative mice from the 3 biological replicates to conduct these analyses.

Figure 5&6: The statistics is not always clear - is Anova an appropriate test? Normal distribution and variance criteria are not met on a few occasions in figure 4&5.

We have responded to the comment on data distribution within Figure 4 below (in Reviewer 4's specific comments). Our statistical analyses within this figure are appropriate.

For Figure 5A, all groups except two were found to have $p > 0.05$ on application of the Shapiro-Wilk test. These were: (1) the WT group for IRF-1/Actin ($W = 0.7459$, $p = 0.0446$) and (2) the KO group for P-STAT1/T-STAT1 ($W = 0.6517$, $p = 0.0028$). There is a strong argument in statistical analysis that introducing different types of testing to a collective may introduce a bias above the benefit of accounting for data distribution. With this in mind, we opted for standardized testing across groups using unpaired t-tests since 90% of groups were $p > 0.05$. However, given the reviewer's concerns, we have repeated these two comparisons using Mann Whitney tests and have confirmed the comparisons are not statistically significant using non-parametric testing: IRF-1/Actin ($p = 0.3095$) and P-STAT1/T-STAT1 ($p = 0.6905$).

A similar assessment was made for Supplementary figure 7A (lane volume) where all groups except one were found to have $p > 0.05$ on application of the Shapiro-Wilk test. This was the WT group for IRF-1/Actin ($W = 0.7059$, $p = 0.0110$). We have repeated the

comparison using a Mann Whitney test and have confirmed the comparison is not statistically significant using non-parametric testing ($p = 0.3095$).

For Figure 5B, a two-way ANOVA with a post hoc multiple comparisons test is the appropriate test as there are two variables that need to be accounted for (genotype x location) and all groups were $p > 0.05$ following the Shapiro-Wilk test.

The manuscript does not contain Figure 6.

Reviewer #2 (Remarks to the Author):

The authors responded to the reviewer's comments, and the manuscript is now somewhat improved. However, there remain some flaws.

In the response to comment#2: the authors provided new data; the expression of claudin2 was increased in the absence of Gbp5. But it remains unclear how GBP5, together with IRF4, regulates the expression of claudin2.

In the response comment#5: The authors did not provide any evidence of how GBP5 and IRF4 interact with each other to maintain the gut homeostasis.

Thus, the findings are interesting. However, the mechanistic insight into how the GBP5/IRF4 axis regulates the intestinal homeostasis is still missing.

We thank the reviewer for their consideration. While we agree that generating mechanistic insight into how GBP5 and IRF4 regulate intestinal homeostasis would be of high interest, we believe this relationship can be explored in future studies to avoid de-emphasizing the main findings from our human randomized clinical trial. However, we appreciate the reviewer's concern and have further limited our referral to IRF4 as a key finding in this study. This includes:

(1) Modification to the title of study, removing mention of IRF4:

“Profiling the colonic mucosal response to fecal microbiota transplantation identifies a role for GBP5 in colitis in humans and mice”

(2) Modification of the conclusive sentences of the Abstract and Discussion to de-emphasize the findings on IRF4:

Abstract:

“The host colonic mucosal response is a strong discriminator of UC remission following FMT, with GBP5, and potentially IRF4, playing a detrimental role in colitis in both humans and mice.”

Discussion:

“GBP5 appears to be a key modulator of colitis in humans and mice.”

Reviewer #3 (Remarks to the Author):

The authors have adequately addressed my concerns.

Reviewer #4 (Remarks to the Author):

Luu et al investigate in their manuscript "Profiling the colonic mucosal response to fecal microbiota transplantation identifies a role for GBP5 and IRF4 in colitis in humans and mice" the transcriptomic response to FMT and perform follow up analysis in animal studies.

GBP5 and IRF4 downregulation was associated with remission. Mice with a defect in GBP5 were more resistant to experimental colitis suggesting a causal link between the therapeutic impact of FMT, regulation of GBP5, and intestinal inflammation.

Comments:

The classification of therapeutic success seems overinflated in Figure 3A and B - 92% correct classification looks optimised since the components substantially overlap.

This is a supervised canonical analysis of principal coordinates using the 78 genes that are significantly regulated in patients that achieved the primary endpoint. It is unclear why the reviewer would feel these genes would not classify that group accurately, albeit with 1 misclassification. The relevance of this analysis is to test how well these genes classify the other groups, particularly the non-responders post-FMT (i.e., Tx8N). Using these genes, the Tx8N group classifies with themselves (n=4), with baseline samples (n=8), or with responders post-FMT (n=1). We provide a possible explanation to these findings within the manuscript (included below for convenience). It is also important to note that CAP analysis is multidimensional. Figure 3B is only a 2D representation of the two axes that explain the most variation and not all variation in the data.

"Of note, the misclassified responder (Figure 3B table) was a patient who experienced endoscopic response but did not achieve the strict endoscopic remission endpoint by week 8, while the non-responder seen to cluster with responders (Figure 3A, right) was a patient who reached the clinical remission endpoint but not the endoscopic endpoint (see ref. 8 for endpoint definitions)."

Please further clarify the location of sampling for biopsies and whether was it the same for all samples.

"Baseline mucosal biopsies were taken from the region identified to be most inflamed in all patients. Longitudinal sampling of patients following FMT or placebo was standardized according to baseline biopsies."

We have further clarified the location of sampling for biopsies as requested. The modified text now reads:

"Baseline mucosal biopsies were taken from the most inflamed segment of the rectosigmoid. Longitudinal sampling of patients following FMT or placebo was standardized according to baseline biopsies."

The authors state "Notably, analysis of the specific cell-types suggested an underpowered difference in the inferred abundances of naïve and memory CD4 T cells (Supplementary Figure 2D),

>an underpowered difference is no difference?

This has been adjusted to simply reflect the results and not assume power. The modified text now reads:

"Notably, analysis of the specific cell-types suggested differences in the inferred abundances of naïve (p=0.065) and memory CD4 T cells (resting, p=0.0095; activated, p=0.05) (Supplementary Figure 2D), which is in line with CD4+ but not CD8+ T cells playing a role in FMT-mediated resolution of C. difficile infection²⁶."

The responders have a dramatically different transcriptomic response > is this difference similar in the placebo response or a mesalazine or anti-TNF response?

We show in Figure 1D that only 3 genes were significantly regulated following placebo treatment. These genes are listed in the supplementary tables. Our trial does not include patients on mesalazine or anti-TNF therapy. We have already de-emphasized the association of the molecular response with FMT treatment, and rather attribute the molecular changes to disease remission.

Statistics- some figures likely need revision:

Figure 4 D&f statistics - unpaired t-test Figure 5b Anova - those groups do not look normal distributed and t test/Anova are likely not appropriate

-> Mann Whitney might be applied - but the significance level will certainly different given the small number and significant overlap

All statistical analyses were performed to the highest standards. This included application of the Shapiro-Wilk test. None of the groups in 4D or 4F reported $p < 0.05$ to indicate a non-parametric test should be applied to the data. While this does not indicate the groups are normally distributed, it does not provide support for application of non-parametric tests. The results of the Shapiro-Wilk test on the original data are below; however, Figure 4F has now been modified to include additional analysis.

Figure 4D

WT group: $W = 0.9092$, $p = 0.2082$

KO group: $W = 0.9117$, $p = 0.1668$

Figure 4F (old data)

WT group: $W = 0.9255$, $p = 0.5461$

KO group: $W = 0.9194$, $p = 0.5008$

REVIEWERS' COMMENTS

Reviewer #2 (Remarks to the Author):

The authors responded to the comments, and the manuscript is now improved.

Reviewer #4 (Remarks to the Author):

The authors have clarified my questions and amended the manuscript when required.

We thank the Reviewers for the time and effort spent reviewing our manuscript.

Reviewer #2 (Remarks to the Author):

The authors responded to the comments, and the manuscript is now improved.

Reviewer #4 (Remarks to the Author):

The authors have clarified my questions and amended the manuscript when required.
